# Go-Browse: Training Web Agents with Structured Exploration

**Apurva Gandhi    Graham Neubig**
Carnegie Mellon University
Pittsburgh, PA
`{apurvag,gneubig}@cs.cmu.edu`

## Abstract

One of the fundamental problems in digital agents is their lack of understanding of their environment. For instance, a web browsing agent may get lost in unfamiliar websites, uncertain what pages must be visited to achieve its goals. To address this, we propose Go-Browse, a method for automatically collecting diverse and realistic web agent data at scale through structured exploration of web environments. Go-Browse achieves efficient exploration by framing data collection as a graph search, enabling reuse of information across exploration episodes. We instantiate our method on the WebArena benchmark, collecting a dataset of 10K successful task-solving trajectories and 40K interaction steps across 100 URLs. Fine-tuning a 7B parameter language model on this dataset achieves a success rate of 21.7% on the WebArena benchmark, beating GPT-4o mini by 2.4% and exceeding current state-of-the-art results for sub-10B parameter models by 2.9%.[1]

## 1 Introduction

Despite their impressive and often superhuman performance in other domains, most pretrained LLMs do not perform well on GUI-based web agent tasks. For instance, on the WebArena benchmark (Zhou et al.) where humans achieve a 78% success rate, frontier models like GPT-4O (OpenAI, 2024a) and GPT-4O-MINI (OpenAI, 2024b) score only 38% and 19% respectively, while a smaller model like QWEN-2.5-7B-INSTRUCT (Yang et al., 2024) scores only 8%. On the other hand, models trained specifically for GUI-based interaction score much better, with CLAUDE-3.7-SONNET (Anthropic, 2025) scoring 45.4% and COMPUTER-USING AGENT (OpenAI, 2025) achieving 58%. This gap suggests that training on agent-specific interaction data is crucial for realizing effective web agents.

But collecting high-quality web agent data presents its own set of challenges. Human-generated trajectories offer one source for quality demonstrations but are notoriously expensive and time-consuming to collect for the vast datasets required. One class of methods tries to automatically scale human-generated data or use humans-in-the-loop in the dataset collection process (Shen et al., 2024; Zhou et al., 2024; Lai et al., 2024). Another line of work attempts to improve scalability further by proposing fully unsupervised and automatic methods for data generation; for example, by generating synthetic demonstrations from wikiHow-style tutorial articles (Ou et al.) or by building an exploration policy that collects data by interacting with websites (Murty et al., 2024a;b).

Among these unsupervised methods, the latter ones that directly explore web environments of interest perform significantly better than those that use indirect and more generic knowledge from the internet (16% (Murty et al., 2024b) vs. 6% (Ou et al.) success rate). This gap underscores a fundamental problem in digital agents: their lack of prior understanding of the environments they are deployed on. Learning from a tutorial or even a human-generated demonstration on how to cancel an ongoing order on Amazon is unlikely to transfer to the myriad of other websites that a web agent may need to interact with. Instead, agents are likely to be more successful if they learn directly from environments they will encounter.

In this work, we introduce GO-BROWSE, a method that automatically collects diverse, realistic, and tailored web agent data through systematic and structured exploration of websites. In particular,

---

[1] We release our code, dataset and models at `https://github.com/ApGa/Go-Browse`.

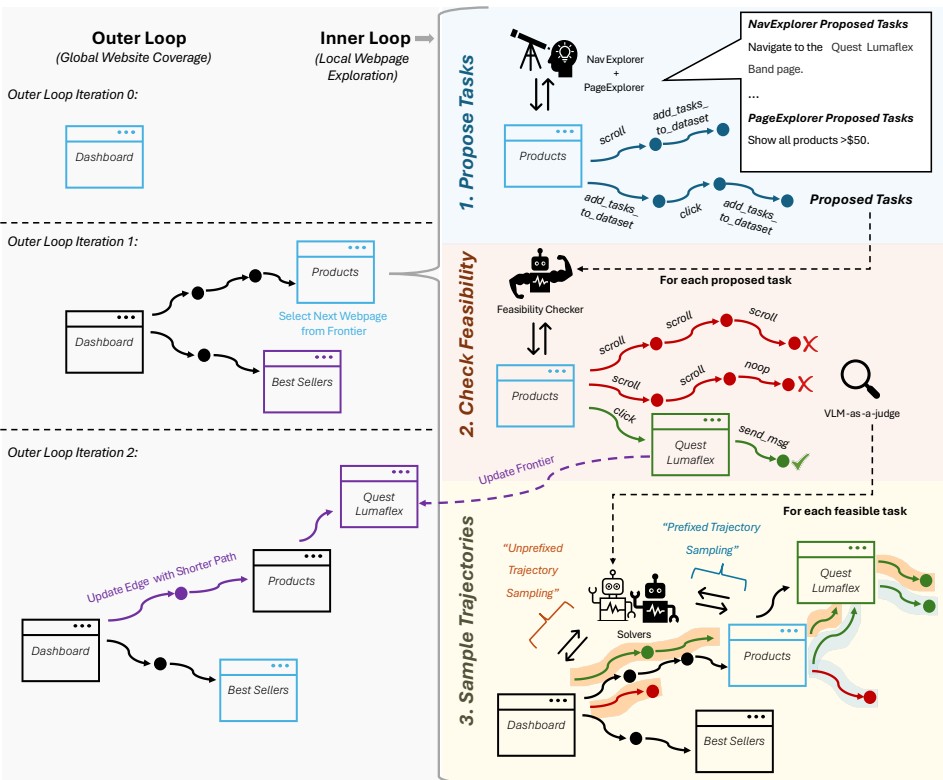

Figure 1: Overview of the GO-BROWSE algorithm for web agent data collection for a website. GO-BROWSE's outer-loop (left) maintains an exploration frontier of discovered but not yet fully explored webpages. GO-BROWSE's inner loop (right) explores each webpage in the frontier by (1) Proposing tasks for that webpage that are grounded in interaction; (2) Checking the feasibility of those tasks; and (3) Sampling trajectories and discovering new webpages by solving feasible tasks.

GO-BROWSE iteratively builds up a graph of previously visited URLs while collecting data. This allows us to reuse information across exploration episodes, resetting new episodes to previously discovered promising webpages, and continuing exploration from there. This improves exploration efficiency over previous unsupervised data collection methods (Murty et al., 2024b;a), which have minimal reuse of information across episodes. Furthermore, resetting to previously discovered webpages allows GO-BROWSE to decouple the challenge of web navigation (finding the correct page) from that of local task solving (performing actions on that page). We demonstrate that this decoupling facilitates a *bootstrapping* effect, enabling even weaker pretrained LLMs to collect higher-quality data, since local task execution is often less demanding than website navigation, which often requires domain-specific knowledge. GO-BROWSE draws inspiration from previous RL works like GO-EXPLORE (Ecoffet et al., 2019; 2021), which used analagous reset-then-explore strategies to solve games like Montezuma's Revenge that have notoriously difficult exploration challenges.

To evaluate GO-BROWSE, we instantiate it on the WebArena benchmark, collecting a dataset of ∼10K successful task-solving trajectories as well as ∼17K unsuccessful trajectories across 100 distinct URLs. Finetuning QWEN-2.5-7B-INSTRUCT on our dataset achieves a task success rate of 21.7% on WebArena. This result surpasses NNETNAV (Murty et al., 2024b), the current state-of-the-art results for sub-10B parameter models by 2.9% and beats GPT-4O-MINI by 2.4%.

## 2 BACKGROUND

### 2.1 LLM WEB AGENTS

Following previous work (Chezelles et al., 2024; Zhou et al.), our web agents are implemented using the ReAct pattern (Yao et al., 2023) where at each timestep $t$ we prompt the LLM with a state $s_t$ and ask it to produce an action $a_t$. Executing $a_t$ in the browser generates a new state $s_{t+1}$.

We include several components in each $s_t$: the task or goal $g$, a flattened accessibility tree representation of the current webpage, a description of the action space, the history of previous actions and any errors encountered when executing the last action. The action space includes primitive operations represented as python functions like `click(id)`, `scroll(dx, dy)`, `type(text)` and `send_msg_to_usr(msg)`, which the agent uses to interact with the browser environment. Each action $a_t$ produced by the LLM consists of a chain-of-thought and a python function call. The complete action space and prompt template are detailed in the Appendix A.

A trajectory $\tau = \{s_1, a_1, s_2, a_2, ..., s_T, a_T\}$ represents a sequence of states and actions taken by the agent in attempting to complete a task. Trajectories terminate either when a maximum horizon length $T$ is reached or when the agent performs a terminal action (such as `send_msg_to_usr(msg)`). For each trajectory, we define a reward model $R(g, \tau) \in \{0, 1\}$ that evaluates task completion success. The binary reward indicates whether the agent successfully completed the specified task ($R(g, \tau) = 1$) or failed ($R(g, \tau) = 0$) by the end of the trajectory. In this work, we leverage the BROWSERGYM (Chezelles et al., 2024) python package to implement our web agent.

## 2.2 EXPLORATION POLICIES FOR DATA COLLECTION

To collect web agent data in an environment through direct interaction, we need to design a method that can explore the environment effectively to gather diverse and high-quality demonstrations, which we refer to as an *exploration policy*. We can classify past work on building exploration policies into two main categories: *interaction-first* and *instruction-first*.

**Interaction-first Exploration Policy.** Interaction-first approaches (Murty et al., 2024a;b), such as NNETNAV roll out an agent (e.g., a prompted pretrained LLM) to explore a website with general exploration instructions (e.g., persona simulation) instead of a concrete task (e.g., *Add Nintendo Switch to cart*). The collected trajectories are then labeled with concrete tasks in retrospect using another prompted LLM, which we call a Labeler ($L$). We call each rollout here an exploration episode. Algorithm 1 provides pseudocode for this process.

A benefit of interaction-first approaches is that the collected trajectories may potentially explore deeper parts of the website that might not be immediately apparent in the initial state. But since each exploration episode operates independently, there's significant redundancy in exploration—agents may often revisit the same parts of websites across different episodes, leading to similar task demonstrations. Additionally, without specific task guidance, agents may spend considerable time collecting trajectories that do not yield interesting or useful tasks.

**Instruction-first Exploration Policy.** Unlike interaction-first approaches, instruction-first exploration policies (Lai et al., 2024; Murty et al., 2024a; Zhou et al., 2024) first generate potential tasks and then attempt to solve them. In this approach, a prompted LLM task proposer $P$ observes a state $s_t$ and generates a set of plausible tasks $\mathcal{G} = \{g_1, g_2, ..., g_K\}$ grounded in the webpage's observed content and functionality. A pretrained policy $A$ then attempts to solve each task $g_i$, generating trajectories $\tau_i = \{s_0, a_0, s_1, a_1, ...\}$ for each proposed task. Finally, a reward model $R(g_i, \tau_i) \in \{0, 1\}$ evaluates whether each trajectory successfully completes its corresponding task, and successful pairs $(g_i, \tau_i)$ where $R(g_i, \tau_i) = 1$ are added to the dataset $\mathcal{D}$. Algorithm 2 provides pseudocode for this approach.

This approach leverages an LLM's prior knowledge to efficiently generate diverse, useful and contextually relevant tasks, but has limitations: proposed tasks are typically limited to the currently observed page, and the LLM may occasionally hallucinate infeasible tasks about unobserved parts of the website. This is because task proposal in these methods is typically anchored to an initial static observation. To address this, works like PAE (Zhou et al., 2024) require screenshots of human demonstrations across the website to gain additional context for task proposal; instead, our work implements the task proposer $P$ with agents that explore and gather their own context automatically.

## 3 GO-BROWSE

We propose GO-BROWSE, which addresses the limitations of past interaction-first and instruction-first approaches. For each website of interest, GO-BROWSE builds a systematic map of previously discovered webpages by treating exploration as a graph traversal problem. It maintains an exploration frontier of discovered but not yet fully explored webpages and progressively explores and expands this frontier by proposing and solving tasks that encourage both local webpage exploration and finding new webpages for global frontier expansion.

| **Algorithm 1** Interaction-first Exploration | **Algorithm 2** Instruction-first Exploration |
|---|---|
| 1: $A \leftarrow$ Agent() | 1: $P \leftarrow$ TaskProposer() |
| 2: $L \leftarrow$ Labeler() | 2: $A \leftarrow$ Agent() |
| 3: $\mathcal{D} \leftarrow \emptyset$ | 3: $R \leftarrow$ RewardModel() |
| 4: | 4: $\mathcal{D} \leftarrow \emptyset$ |
| 5: **for** website $W \in \mathcal{W}$ **do** | 5: **for** website $W \in \mathcal{W}$ **do** |
| 6:     **for** $1 \ldots N$ iterations **do** | 6:     $s_0 \leftarrow$ InitialState($W$) |
| 7:         $s_0 \leftarrow$ InitialState($W$) | 7:     $\mathcal{G} \leftarrow P(s_0,)$ |
| 8:         $g \leftarrow$ Exploration instructions. | 8:     **for** task $g \in \mathcal{G}$ **do** |
| 9:         $\tau \leftarrow$ SampleTrajectory($A, s_0, g$) | 9:         $\tau \leftarrow$ SampleTrajectory($A, s_0, g$) |
| 10:         $\mathcal{D}_\tau \leftarrow L(\tau)$ | 10:         **if** $R(g, \tau) = 1$ **then** |
| 11:         $\mathcal{D} \leftarrow \mathcal{D} \cup \mathcal{D}_\tau$ | 11:             $\mathcal{D} \leftarrow \mathcal{D} \cup \{(g, \tau)\}$ |
| 12:     **end for** | 12:         **end if** |
| 13: **end for** | 13:     **end for** |
| 14: | 14: **end for** |
| 15: **return** $\mathcal{D}$ | 15: **return** $\mathcal{D}$ |

Figure 2: Comparison of common styles of exploration policies for web agent data collection.

Fig. 1 illustrates the GO-BROWSE algorithm and Algorithm 3 provides pseudocode. Specifically, GO-BROWSE builds up a graph $G = (\mathcal{V}, \mathcal{E})$, where nodes $v \in \mathcal{V}$ are unique URLs and edges $e \in \mathcal{E}$ are trajectories between them. As shown in Fig. 1, its outer loop (left) resembles graph traversal (e.g., breadth-first search), while an inner loop (right) resembles instruction-first exploration.

In each outer loop iteration, GO-BROWSE first selects a webpage $v$ from the frontier and then performs the following inner loop to collect data and explore $v$: (1) Propose navigational and local tasks for $v$ using the `NavExplorer` and `PageExplorer` modules, (2) Check feasibility of proposed tasks with the `FeasibilityChecker` module, and (3) Sample trajectories by solving feasible tasks with the `Solvers` module. We describe these modules below. GO-BROWSE's outer loop enforces global coverage of the website while the inner loop thoroughly explores each discovered webpage.

**NavExplorer: Frontier Expansion and Navigational Task Collection.** The `NavExplorer` module is responsible for proposing navigational tasks to webpages that neighbor the current webpage $v$ in the graph. This is similar to the `TaskProposer` module in instruction-first approaches, but instead of just asking the LLM to propose tasks from a static observation, we instead implement `NavExplorer` as a web agent itself. We instruct it with a goal $g$ to find neighboring webpages through interaction with the current webpage, and propose navigational tasks to reach them. We do the latter by extending the action space of the `NavExplorer` agent with an `add_tasks_to_dataset(tasks: tuple[str])` function. Designing `NavExplorer` as a web agent empowers it to perform its own purposeful exploration and ground proposed tasks on dynamically obtained observations. To keep the exploration process efficient and prioritize nodes added to the frontier, `NavExplorer` is asked to prioritize adding tasks for navigating to new webpages that are likely to have common and useful tasks that a user might want to perform. Full prompts for the `NavExplorer` modules are provided in Appendix A.2.

**PageExplorer: Local Page Exploration and Task Collection.** The `PageExplorer` is similar to the `NavExplorer`, except that it is responsible for proposing tasks local to the current webpage $v$. It does so by asking an LLM to generate a set of plausible tasks that a user may want to perform on the current webpage (prompts in Appendix A.2) The tasks generated by the `PageExplorer` help generate training data that thoroughly explore the functionality of each webpage.

**FeasibilityChecker: Task Filtering and Trajectory Sampling.** The `FeasibilityChecker` module filters the tasks proposed by the previous two modules by (1) using a strong pretrained LLM agent (e.g., computer-use trained LLM) to try and solve each task, and (2) using a pretrained VLM-as-a-judge to check if the sampled trajectory solves the task. We sample up to $N_{max}$ trajectories, stopping if we sample a success. Proposed tasks with at least one successful trajectory are considered feasible and kept in the dataset along with their corresponding trajectories, while the rest are discarded.

**Solvers: Prefixed and Unprefixed Sampling.** The `Solvers` sample additional trajectories for the filtered, feasible tasks, but can use cheaper models to sample a larger number of trajectories. Additionally, `Solvers` perform a mix of *prefixed* and *unprefixed* sampling. In prefixed sampling,

---

**Algorithm 3** Go-Browse

---

1: Initialize Dataset $\mathcal{D} \leftarrow \emptyset$, Graph $G = (\mathcal{V}, \mathcal{E})$, Frontier $F \leftarrow \emptyset$
2: Initialize Modules: NavExplorer, PageExplorer, FeasibilityChecker, Solvers, RewardModel $R$
3: **for** each website $W_i \in \mathcal{W}$ **do**
4: $\quad$ $v_{\text{root}} \leftarrow$ GetRootURL($W_i$); Add $v_{\text{root}}$ to $F$ and $\mathcal{V}$
5: $\quad$ **while** $F$ is not empty **do**
6: $\quad\quad$ $v \leftarrow$ SelectAndRemoveFromFrontier($F$)
7: $\quad\quad$ $s_v \leftarrow$ GetCurrentState($v$)
$\quad\quad\quad\quad\quad\quad\quad\quad\quad\quad\quad\quad\quad\quad\quad\quad\quad\quad\quad\quad$ ▷ Propose navigation and local tasks
8: $\quad\quad$ $\mathcal{G}_{\text{nav}} \leftarrow$ NavExplorer.propose_tasks($s_v$)
9: $\quad\quad$ $\mathcal{G}_{\text{local}} \leftarrow$ PageExplorer.propose_tasks($s_v$)
10: $\quad\quad$ $\mathcal{G}_{\text{proposed}} \leftarrow \mathcal{G}_{\text{nav}} \cup \mathcal{G}_{\text{local}}$
11: $\quad\quad$ $\mathcal{G}_{\text{feasible}} \leftarrow \emptyset$
$\quad\quad\quad\quad\quad\quad\quad\quad\quad\quad\quad\quad$ ▷ Filter for feasible tasks and collect initial trajectories
12: $\quad\quad$ **for** task $g \in \mathcal{G}_{\text{proposed}}$ **do**
13: $\quad\quad\quad$ (is_feasible, $\tau_{\text{fc}}$, $v_{\text{new}}$) $\leftarrow$ FeasibilityChecker.check_and_collect($g, s_v, R, N_{max}$)
14: $\quad\quad\quad$ **if** is_feasible **then**
15: $\quad\quad\quad\quad$ Add $(g, \tau_{\text{fc}})$ to $\mathcal{D}$; Add $g$ to $\mathcal{G}_{\text{feasible}}$
16: $\quad\quad\quad\quad$ **if** $v_{\text{new}}$ is a new discovered URL **then**
17: $\quad\quad\quad\quad\quad$ Add $v_{\text{new}}$ to $\mathcal{V}$ and $F$; Add new edges to $\mathcal{E}$
18: $\quad\quad\quad\quad$ **end if**
19: $\quad\quad\quad$ **end if**
20: $\quad\quad$ **end for**
$\quad\quad\quad\quad\quad\quad\quad\quad\quad\quad$ ▷ Sample additional prefixed and unprefixed trajectories
21: $\quad\quad$ **for** feasible task $g \in \mathcal{G}_{\text{feasible}}$ **do**
22: $\quad\quad\quad$ $\mathcal{T}_{\text{prefixed}} \leftarrow$ Solvers.sample($g, s_v, R$, prefixed=True)
23: $\quad\quad\quad$ $\mathcal{D} \leftarrow \mathcal{D} \cup \{(g, \tau) \mid \tau \in \mathcal{T}_{\text{prefixed}}\}$
24: $\quad\quad\quad$ $s_{\text{root}} \leftarrow$ GetState(GetRootURL($W_i$))
25: $\quad\quad\quad$ $\mathcal{T}_{\text{unprefixed}} \leftarrow$ Solvers.sample($g, s_{\text{root}}, R$, prefixed=False)
26: $\quad\quad\quad$ $\mathcal{D} \leftarrow \mathcal{D} \cup \{(g, \tau) \mid \tau \in \mathcal{T}_{\text{unprefixed}}\}$
27: $\quad\quad$ **end for**
28: $\quad$ **end while**
29: **end for**
30:
31: **return** $\mathcal{D}$

---

the agent tries to solve $g$ starting from the current webpage $v$, while in unprefixed sampling, the agent has to solve $g$ starting from the root node of the webpage (e.g., usually the homepage or dashboard). Prefixed sampling makes the agent's job easier by decoupling navigation (finding the webpage) from task solving locally on that webpage. As we discuss in Section 6, prefixed sampling has higher success rates, letting us bootstrap from even weaker pretrained models. Still, it is useful to sample unprefixed trajectories to instill long-horizon task-solving and exploratory behaviors in the agent.

**Relation to Instruction-First and Interaction-First Approaches.** We can think of GO-BROWSE's inner-loop interaction between the `NavExplorer`, `PageExplorer` and `FeasibilityChecker` as a form of *instruction-first* exploration. But unlike typical instruction-first approaches that only start at the root node (homepage, dashboard, etc.) of the website, GO-BROWSE's inner-loop is initialized with new pages from the frontier at every iteration. This addresses the localized exploration of instruction-first approaches by enforcing global website coverage. Furthermore, by using web agents for task proposal, GO-BROWSE enables more grounded task proposal based on real observations. GO-BROWSE also addresses the exploration efficiency limitation of interaction-first approaches by reusing information from past episodes. Since each iteration of the outer-loop resets exploration to a previously discovered webpage, GO-BROWSE can reduce redundancy and instead spend more budget on exploring novel parts of the website.

## 4  DATA COLLECTION

We collect a dataset (GO-BROWSE-WA) by running GO-BROWSE on the WebArena benchmark (Zhou et al.), which consists of 5 self-hosted websites, representing clones of common domains: Shopping Admin (CMS), Shopping, Reddit, Gitlab, and Map. We explore 20 different URLs for each

of the five domains, collecting tasks across 100 distinct URLs. While our experiments focus on these domains, we note that the same data collection pipeline can be run on other websites of interest.

For the `NavExplorer` we perform up to 15 steps of interaction with CLAUDE-3.7-SONNET (Anthropic, 2025). For the `PageExplorer` we perform up to 20 steps with GPT-4O (OpenAI, 2024a) and 10 steps with CLAUDE-3.7-SONNET. Appendix B.1 provides analysis of some of the complementary differences in behavior between these models. The `FeasibilityChecker` uses CLAUDE-3.7-SONNET to try solving proposed tasks, with a maximum of 3 tries, and uses a GPT-4O-based *"VLM-as-a-judge"* reward model (adopted from Pan et al. (2024); Wang et al. (2025)). We keep a maximum of 30 feasible tasks per URL. For the `Solvers` we use GPT-4O-MINI and QWEN-2.5-7B-INSTRUCT. Task solving is limited to a maximum horizon length of 10 steps. The `Solvers` sample 2 prefixed trajectories and 2 unprefixed ones. For all interaction steps in the dataset collection process we use a temperature of 0.7. Overall, GO-BROWSE-WA took $\sim$ \$975.57 to collect; for a detailed cost analysis, see Appendix B.2.

Table 1 shows the composition statistics of the GO-BROWSE-WA dataset. The dataset contains similar proportions of successful trajectories from each model we use to sample trajectories (Fig. 3). While here we only finetune on the success steps, we release all steps in our dataset, including failures. The dataset also includes alternate representations of webpage observations (accessibility tree, HTML, and screenshots); although, we only use the accessibility tree for our finetuning experiments.

Table 1: Dataset statistics on the 5 WebArena domains (20 pages explored/domain).

|              | Success | Failure | Total   |
| ------------ | ------- | ------- | ------- |
| Trajectories | 9,504   | 17,245  | 26,749  |
| Steps        | 39,339  | 157,123 | 196,462 |
| Unique tasks |         | 3,422   |         |

| Qwen-2.5 7B Instruct | GPT-4o Mini | Claude-3.7 Sonnet |
| -------------------- | ----------- | ----------------- |
| 29.5%                | 36.6%       | 33.9%             |

Figure 3: Proportion of successful trajectories from each model in the Go-Browse-WA dataset.

## 5 FINE-TUNING SETUP AND RESULTS

For our experiments, we train QWEN-2.5-7B-INSTRUCT by performing supervised finetuning on only the success trajectories of our dataset. Finetuning hyperparameters are provided in Appendix C. For fair comparison, we also train a model using the same parameters on the NNETNAV-WA dataset (Murty et al., 2024b) which consists of 45K interaction steps across the 5 WebArena domains.

We benchmark the finetuned models on the 812 WebArena benchmark using BROWSERGYM (Chezelles et al., 2024). Correctness of each task is evaluated using task-specific reward functions provided by WebArena. We use a temperature of 0 for the models when benchmarking.

Table 3 shows the success rates of the finetuned models on WebArena as well as other pretrained models. Our model, GO-BROWSE-7B, achieves a success rate of 21.7% overall on the WebArena tasks, outperforming the other models in the table. The GO-BROWSE-7B model outperforms the pretrained QWEN-2.5-7B-INSTRUCT model by 13.4% and the finetuned NNETNAV-7B model by 2.9%. Notably, GO-BROWSE-7B also outperforms GPT-4O-MINI by 2.4%. Appendix B.3 provides results of performing statistical significance testing with paired bootstrap tests.

Looking at individual domains, GO-BROWSE-7B scores higher than GPT-4O-MINI and NNETNAV-7B in all domains except for Gitlab. Notably, GO-BROWSE-7B beats NNETNAV-7B by 11% on the Shopping Admin domain and 7% on the Reddit domain.

We also evaluate our models on Online-Mind2Web (OM2W) (Xue et al., 2025), an out-of-domain benchmark with 300 tasks across 136 live websites. GO-BROWSE-7B still maintains a lead over NNETNAV-7B even in this generalization experiment, though—as expected—models perform worse in this setting compared to the in-domain WebArena. GPT-4O-MINI also scores much worse on OM2W. Appendix B.4 includes further analysis and discussion,

| Model        | SR (%) |
| ------------ | ------ |
| NNetNav-7B   | 4.00   |
| Go-Browse-7B | 5.33   |
| GPT-4o-mini  | 9.33   |

Table 2: Online-M2W results.

showing that on OM2W websites that are similar to WebArena (In-Domain-Adjacent websites), GO-BROWSE-7B approaches the performance of GPT-4O-MINI (<1% SR difference) while still performing better than NNETNAV-7B by 3%.

Table 3: Success rates on WebArena tasks. Bold indicates the best result in each category.

| Model | Overall (%) | Admin (%) | Shopping (%) | Reddit (%) | Gitlab (%) | Map (%) |
|---|---|---|---|---|---|---|
| *Closed Models.* | | | | | | |
| GPT-4O-MINI | 19.3 | 19.2 | 19.3 | 21.1 | 20.9 | 15.6 |
| GPT-4O | 37.6 | 35.7 | 32.3 | 50.9 | 36.7 | 37.5 |
| CLAUDE-3.7-SONNET | **45.4** | **37.4** | **37.0** | **58.8** | **52.0** | **47.7** |
| *Open-weights 7B Models.* | | | | | | |
| QWEN-2.5-7B-INSTRUCT | 8.3 | 7.1 | 9.4 | 7.9 | 8.7 | 7.8 |
| NNETNAV-7B | 18.8 | 14.3 | 20.3 | 23.7 | **19.9** | 17.2 |
| GO-BROWSE-7B | **21.7** | **25.3** | **22.4** | **30.7** | 15.3 | **17.9** |

# 6 ANALYSIS

**Go-Browse Generates Diverse Tasks.** Fig. 4 compares the distribution of tasks in the NNETNAV-WA and GO-BROWSE-WA datasets. We follow Murty et al. (2024b) in using GPT-4O-MINI to cluster dataset tasks into higher-level intent categories. We can see that NNETNAV shows a tendency to have larger wedges in its task distribution, indicating redundancy in exploration since each episode is independent. This pattern is particularly evident in domains that are challenging to navigate, such as Shopping Admin, because a larger number of episodes will navigate to the same easy-to-find webpages, leading to similar tasks; harder to find webpages, even if discovered by one episode, may rarely be revisited in a future episode. GO-BROWSE addresses this issue by resetting to previously encountered webpages. Even if a page is difficult to navigate to, once it is discovered, GO-BROWSE makes sure it is thoroughly explored in future episodes.

GO-BROWSE-WA also exhibits a more balanced distribution across domains; NNETNAV contains a disproportionately large number of Gitlab tasks and relatively few Reddit tasks. These observations align with our model performance results in Table 3, where NNETNAV-7B only outperforms GO-BROWSE-7B on the Gitlab domain, and performs notably worse on Shopping Admin and Reddit.

**Go-Browse's Successful Trajectories Go Deeper.** Fig. 5 plots how deep into the website the finetuned models go when solving tasks. On the left, when considering all trajectories, we see that both GO-BROWSE and NNETNAV behave similarly, but on trajectories where only GO-BROWSE was successful (middle), we see that the depth distribution is more right-skewed, suggesting that GO-BROWSE owes some of its wins to its tendency to solve longer-horizon tasks. If we look at the NNETNAV-only successful trajectories (right), we again see no significant difference in behavior, showing that going deeper is a unique characteristic of GO-BROWSE's successes.

Table 4 shows URL patterns with the largest difference in success trajectory visits between GO-BROWSE and NNETNAV. GO-BROWSE's successes more frequently involve navigating to deeper URLs, such as editing specific product attributes or viewing particular order details. Notably, GO-

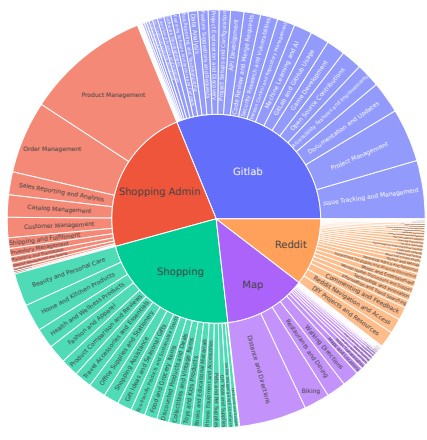

(a) NNETNAV-WA Task Distribution

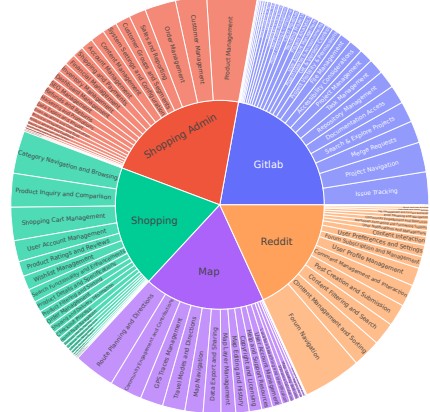

(b) GO-BROWSE-WA Task Distribution

Figure 4: Task diversity of the Go-Browse and NNetNav datasets. Zoom to read sub-task labels.

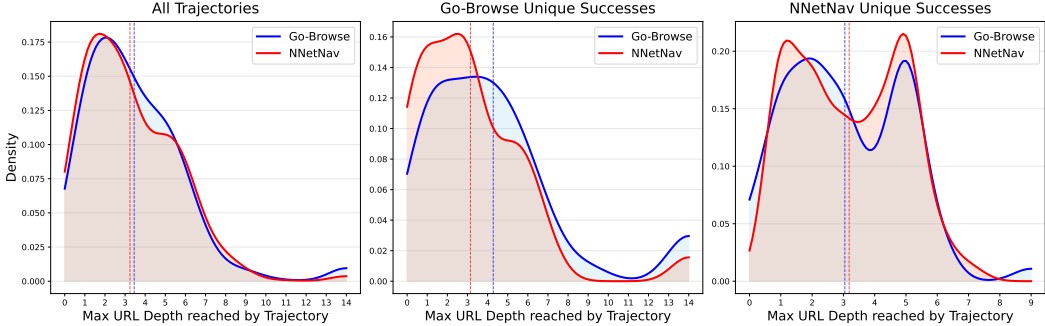

Figure 5: Distributions of maximum URL path lengths (depth) achieved by trajectories across all trajectories (left), trajectories where only Go-Browse was successful (middle), and trajectories where only NNetNav was successful (right). Go-Browse owes some of its wins to its tendency to go deeper. Note, depth is calculated as the number segments in the URL path.

Table 4: Top URLs by difference in visit count (GO-BROWSE (GB) vs. NNETNAV (NN)).

| More Visits By | URL | GB Visits | NN Visits | Diff. | Depth |
|---|---|---|---|---|---|
| GB | `<shopping_admin>/catalog/product/edit/id/{id}/` | 10 | 1 | 9 | 5 |
| | `<reddit>/search?query={query}?q=%7Bquery%7D` | 7 | 0 | 7 | 2 |
| | `<shopping_admin>/catalog/.../configurable/store/{id}/back/edit/` | 5 | 0 | 5 | 12 |
| | `<shopping_admin>/sales/order//view/order_id/{id}//{id}/` | 6 | 2 | 4 | 6 |
| | `<reddit>/user/{user}/edit_biography` | 5 | 0 | 5 | 3 |
| NN | `<gitlab>/projects/new` | 2 | 6 | 4 | 2 |
| | `<gitlab>/projects/new#blank_project` | 2 | 5 | 3 | 2 |
| | `<gitlab>/{user}/{repo}/-/commits/main` | 2 | 5 | 3 | 5 |
| | `<gitlab>/{user}/{repo}/-/forks/new` | 1 | 4 | 3 | 5 |
| | `<reddit>/forums/by_submissions/{id}` | 0 | 3 | 3 | 3 |

BROWSE exhibits significantly higher visit counts to these deeper URLs, including several that NNETNAV never successfully visited. For instance, GO-BROWSE visited URLs for editing product attributes and searching Reddit 9 and 7 more times respectively, with NNETNAV having 1 or 0 visits to these. Conversely, while NNETNAV more frequently visits URLs related to creating new projects or forks in Gitlab, the difference in visitation counts is comparatively smaller. It is also interesting to observe their differing strategies for Reddit: GO-BROWSE tends to use the more direct search functionality, whereas NNETNAV attempts to find forums by navigating to the `by_submissions` page. This aligns with our earlier finding that GO-BROWSE tends to succeed on tasks requiring deeper navigation.

**Prefixed Sampling Bootstraps Weaker Models.** Fig. 6 plots success rates of prefixed and unprefixed sampling against the depth of the node (URL) on which the tasks were sourced. Overall, prefixed sampling leads to higher success rates. The difference is especially pronounced as depth increases: it is harder to find deeper nodes again when starting from the root. The difference is also especially apparent for weaker models like QWEN-2.5-7B-INSTRUCT. Prefixed sampling thus allows us to bootstrap from weaker pretrained models, enabling creation of higher quality data compared to what pretrained models can generate on their own.

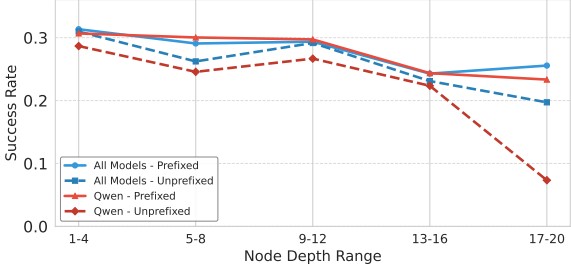

Figure 6: Prefixed sampling leads to higher success rates, especially on deeper nodes, particularly for weaker models like QWEN-2.5-7B-INSTRUCT. Node depth is shortest trajectory length to reach a node from the root node, calculated using Dijkstra's algorithm.

**FeasibilityChecker Improves Exploration Efficiency.** During data collection, 403 tasks were filtered out by the `FeasibilityChecker`. This corresponds to a reduction of 3.2K trajectory rollouts ($\sim 29.4k$ steps). This is a 13% reduction in steps for the same amount of positive data.

**Go-Browse's Outer Loop Improves Website Coverage.** We analyze the effect of GO-BROWSE's outer loop by varying how many discovered URLs we reset to when proposing tasks, while fixing the total proposed tasks at 30 per domain. The 1/30 case corresponds to Go-Browse without its outer loop (proposing 30 tasks for just a single URL per domain). As the number of reset nodes increases, the number of unique URLs visited grows steadily, demonstrating that Go-Browse's outer loop is critical for broader website coverage and more representative data.

| (Resets / Tasks) | Unique URLs |
|---|---|
| (1 / 30) | 183 |
| (5 / 6) | 214 |
| (15 / 2) | 260 |

Table 5: Unique URLs visited across 5 domains with varying # of resets.

## 7 RELATED WORK

**LLM Web Agents.** We build on multiple works on LLM web agents like the WebArena benchmark (Zhou et al.) and BrowserGym (Chezelles et al., 2024) which both provide infrastructure for building and evaluating web agents and also provide an action space and observation features that we leverage in ReAct-based web agents (Yao et al., 2023). There is also a line of works that augment the action space (e.g., with developer APIs or self-learned workflows) (Song et al., 2024; Wang et al., 2025; Zheng et al., 2025) or perform a form of in-context learning by extending observations with additional context from past interactions (Wang et al., 2024; Murty et al., 2024a; Wang et al., 2025). Our work instead improves the base agentic capabilities of LLMs while keeping the scaffolding minimal.

**Synthetic Data Generation for Web Agents.** Past works on generating synthetic data for web agents have focused on either generating data from static indirect knowledge on the internet (e.g., tutorial articles) (Ou et al.) or by logging direct interactions with websites (Shen et al., 2024; Lai et al., 2024; Murty et al., 2024a;b; Zhou et al., 2024). Among the latter methods, interaction-first methods (Murty et al., 2024b) seem to work unsupervised, while instruction-first methods (Zhou et al., 2024; Shen et al., 2024; Lai et al., 2024) have typically required a human-in-the-loop to provide additional context. In our work, we build an unsupervised instruction-first method that can gather its own context via exploration. We also note more recent, concurrent instruction-first/hybrid methods for unsupervised collection of web agent data (Pahuja et al., 2025; Sun et al., 2024; Trabucco et al., 2025). A key difference is GO-BROWSE's focus on deeply exploring websites by explicitly building and leveraging its own web graph. This helps GO-BROWSE collect high-quality and high-coverage data that enables training a state-of-the-art model for WebArena.

**Exploration Methods in Reinforcement Learning.** There is also a rich line of work from the RL community on improving exploration in agents (Bellemare et al., 2016; Pathak et al., 2017; Burda et al., 2019; Ecoffet et al., 2019; 2021). Of these, our method takes the most inspiration from Go-Explore (Ecoffet et al., 2019; 2021) which uses analogous reset-then-explore strategies to share information across episodes to improve exploration efficiency in the context of Atari games.

## 8 CONCLUSION AND LIMITATIONS

In this work, we propose GO-BROWSE, a fully unsupervised and scalable method for collecting web agent data through structured exploration of websites. We release, GO-BROWSE-WA, a dataset 9.5K successful and 39K unsuccessful task solving trajectories obtained while exploring the WebArena environments. We show that simple supervised finetuning of 7B parameter LLM on this dataset leads to significant improvements in success rates of web agents over the the previous state-of-the-art for sub-10B parameter models and also beats GPT-4O-MINI. We thoroughly analyze the characteristics of our dataset and trained models, showing that GO-BROWSE-WA contains high-quality and diverse trajectories that lead to models that are able to better and more deeply navigate explored websites.

There are a number of limitations in our current experimental setup that open promising avenues for future research. Expanding data collection to a broader range of websites beyond the five WebArena domains would allow us to generate even larger datasets. While our current method achieves strong results using a 7B model trained on only successful trajectories, incorporating the signal from the

39K unsuccessful trajectories by exploring alternative training objectives (e.g., RL-based objectives) and scaling up model size may unlock even greater performance improvements. Finally, while using LLMs helps us scale training data, this risks introducing biases from models and prompts that may propagate to agent behavior, requiring careful auditing and mitigation before deployment.

## REPRODUCIBILITY STATEMENT

We release our code, dataset and models at `https://github.com/ApGa/Go-Browse` with documentation on how to reproduce our dataset collection and experiment runs. Additionally, we describe dataset collection hyperparameters in Section 4 and finetuning hyperparameters in Section 5 and Appendix C.

## ACKNOWLEDGEMENTS

We would like to thank Aviral Kumar, Vijay Viswanathan, Yueqi Song and Zora Zhiruo Wang for insightful discussions and feedback. We also thank the CMU Foundation and Language Model (FLAME) Center for access to their compute cluster. This work is supported in part by Amazon.

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

# A  WEB AGENT IMPLEMENTATION DETAILS

## A.1  WEB AGENT ACTION SPACE

Table 6 shows the action space used for web agent experiments, adopted from the BrowserGym framework Drouin et al. (2024).

| Action Type | Description |
|---|---|
| `noop(wait_ms)` | Do nothing for specified time. |
| `click(elem)` | Click at an element. |
| `hover(elem)` | Hover on an element. |
| `fill(elem, value)` | Type into an element. |
| `keyboard_press(key_comb)` | Press a key combination. |
| `scroll(x, y)` | Scroll horizontally or vertically. |
| `select_option(elem, options)` | Select one or multiple options. |
| `goto(url)` | Navigate to a url. |
| `go_back()` | Navigate to the previous page. |
| `go_forward()` | Navigate to the next page. |
| `new_tab()` | Open a new tab. |
| `tab_close()` | Close the current tab. |
| `tab_focus(index)` | Bring tab to front. |
| `send_msg_to_user(text)` | Send a message to the user. |
| `report_infeasible(reason)` | Notify user that instructions are infeasible. |

Table 6: Web Agent action space.

## A.2  PROMPTS FOR LM COMPONENTS

**Prompt Template for Web Agents**

# Instructions
You are a UI Assistant, your goal is to help the user perform tasks using a web browser. Review the instructions from the user, the current state of the page and all other information to find the best possible next action to accomplish your goal. Your answer will be interpreted and executed by a program, make sure to follow the formatting instructions.

# Goal **{Goal}**

#Action Space
**{Action space description from Table 6}**

Here are examples of actions with chain-of-thought reasoning:
{"thought": "I now need to click on the Submit button to send the form. I will use the click action on the button, which has bid 12.", "action": "click('12')"}
{"thought": "I found the information requested by the user, I will send it to the chat.", "action": "send_msg_to_user('The price for a 15 inch laptop is 1499 USD.')"}
{"thought": "I have finished navigating to the Products page. I will inform the user that I have completed the task.", "action": "send_msg_to_user('I have finished navigating to the Products page.')"}

# Current Accessibility Tree
**{Axtree Text}**

# Error Message from Last Action
**{Last Action Error}**

# History of Past Actions
**{Past Actions}**

# Next Action
You will now think step by step and produce your next best action. Reflect on your past actions, any resulting error message, the current state of the page before deciding on your next action. Provide your output as a single json with a thought and an action. All reasoning must be contained within the thought key of the json output, and only a single action must be provided for the action key. Future actions will be taken subsequently. If you have finished performing the request, send a message to the user in a concise and to the point manner.

## Goal for NavExplorer

I am trying to collect a dataset to train a better web browser agent that can perform actions for users in a web browser. For this, we are particularly interested to collect **navigation tasks** that are feasible to perform from the current web page.
Navigation tasks are tasks requiring navigating to a specific page.

Collect navigation tasks that require navigating to another webpage from this current page. You may click on links to try finding other interesting pages to collect tasks from. But if you do navigate to another page, instead of collecting tasks on that page, make sure to navigate back to the previous page using 'go_back' or 'goto'. We will collect tasks from these new pages later. When collecting navigation tasks, prioritize those that would likely have interesting/useful tasks on them over ones that likely won't give many useful tasks to collect.

As you are exploring, you can add navigation tasks to the dataset using the 'add_tasks_to_dataset' function.

When you are done exploring the current page, send a message to the user using 'send_msg_to_user' confirming this.

Be sure to prioritize adding navigation tasks to pages that a typical user of this web page would most often want to navigate to, over niche pages that the typical user would rarely frequent.

**Important** Remember that if you are successful at navigating to a new page, you should add a corresponding task to the dataset as your next action before finding new pages.

## Goal for PageExplorer

I am trying to collect a dataset to train a better web browser agent that can perform actions for users in a web browser. For this, I need to first collect tasks that are feasible to perform on the current web page. The tasks should be concrete (e.g., on an amazon product page for product X, an appropriate task could be "Leave a positive review for X" or on a maps website a task could be "Show me driving directions from X to Y." where X and Y are specific locations).
You may explore by performing actions on this web page if that helps to determine concrete tasks that are feasible.

Find the tasks that are possible to perform on the current web page itself, without have to navigate to other links/urls. Though, you may find it helpful to navigate through menus on this page to get a better idea of what types of tasks are feasible. If you accidentally go to a new url while trying to navigate items on the page, you can go back to the previous page using the 'go_back' function.

Tasks are usually of three types:
1. Information seeking: The user wants to obtain certain information from the webpage, such as the information of a product, reviews, map info, comparison of map routes, etc.
2. Site navigation: The user wants to navigate to a specific page.
3. Content modification: The user wants to modify the content of a webpage or configuration.

Be as specific as you can while creating tasks. The web agent may start from a different web page when asked to complete the task and so may not have the current page context to understand the task. So, for example, avoid creating generic tasks like "Add item to cart" or "Print receipt for this order." Instead you want to create specific tasks like "Add a Sony PS5 to cart" or "Print a receipt for Martha Jone's order of the Nike Velocity Sweatpants from May 21, 2021"

I recommend the following order to collecting tasks:
1. First look for information seeking/extraction tasks that can be answered simply using information on the current page, requiring no additional actions.
2. Collect navigation tasks that require navigating to another webpage from this current page. You may click on links to try finding other interesting pages to collect tasks from. But if you do navigate to another page, instead of collecting tasks on that page, make sure to navigate back to the previous page using 'go_back'. We will collect tasks from these new pages later. When collecting navigation tasks, prioritize those that would likely have interesting/useful tasks on them over ones that likely won't give many useful tasks to collect.
3. Finally, you can try to find content modification tasks on the current page that require performing actions on the current page itself.

As you are exploring the page, you may find it helpful to click on buttons, links, and other elements on the page to see if they reveal any additional information or options that could lead to new tasks. You can also hover over elements to see if they provide any tooltips or additional context.

**Important**:
When collecting tasks, focus more on the common tasks that a typical user of this webpage would want to perform. Avoid niche tasks that are unlikely to be relevant to the typical user of this website.
For most common styles of tasks, it may be useful to include a few variants or related tasks to help the web agent learn frequently used skills.

As you are exploring, you can add tasks to the dataset using the 'add_tasks_to_dataset' function.

When you are done exploring, send a message to the user using 'send_msg_to_user' confirming this.

---

**Prompt for VLM-as-a-judge Reward Model**

You are an expert in evaluating the performance of a web navigation agent. The agent is designed to help a human user navigate a website to complete a task. Given the user's intent, the agent's action history, the final state of the webpage, and the agent's response to the user, your goal is to decide whether the agent's execution is successful or not. Please be careful of each detail and strict about the evaluation process.

There are three types of tasks:

1. Information seeking: The user wants to obtain certain information from the webpage, such as the information of a product, reviews, map info, comparison of map routes, etc. The bot's response must contain the information the user wants, or explicitly state that the information is not available. Otherwise, e.g. the bot encounters an exception and respond with the error content, the task is considered a failure. Besides, be careful about the sufficiency of the agent's actions. For example, when asked to list the top-searched items in a shop, the agent should order the items by the number of searches, and then return the top items. If the ordering action is missing, the task is likely to fail.

2. Site navigation: The user wants to navigate to a specific page. Carefully examine the bot's action history and the final state of the webpage to determine whether the bot successfully completes the task. No need to consider the bot's response.

> 3. Content modification: The user wants to modify the content of a webpage or configuration. Carefully examine the bot's action history and the final state of the webpage to determine whether the bot successfully completes the task. No need to consider the bot's response.
>
> User Intent: **{Goal}**
>
> Action History:
> **{Last Actions}**
>
> The final state of the webpage provided as an accessibility tree:
> **{Axtree Text}**
>
> The last snapshot of the web page is shown in the image.
> **{Screenshot}**

# B  ADDITIONAL ANALYSES

## B.1  DESIGN CHOICES FOR TASK PROPOSAL

### B.1.1  GPT-4O VS. CLAUDE-3.7-SONNET FOR PAGEEXPLORER TASK PROPOSAL

To understand how task proposal behavior differs based on model choice, we tag proposed Page-Explorer tasks using an LLM as navigational (`Nav`), information-seeking (`Info`), or state/content-modifying (`Mod`) tasks (the same three categories mentioned in the PageExplorer goal). We also perform clustering of these tasks to measure diversity, similar to Section 6.

The models differ in task proposal behavior as shown in Table 7: (1) CLAUDE-3.7-SONNET proposes almost almost double the number of tasks with half the max step budget; (2) GPT-4O generates a more diverse set of tasks for the quantity proposed, especially Mod tasks, where GPT-4O has many more task clusters.

The efficiency of Claude allows us to give it a smaller max step budget when used as a PageExplorer. On the other hand, GPT-4O 's diversity of `Mod` tasks justifies using it as well to complement Claude.

Table 7: Comparison of GPT-4O and CLAUDE-3.7-SONNET for PageExplorer agents.

| Model | # Tasks | | | # Clusters | | | Max # Steps (Per Node) |
|---|---|---|---|---|---|---|---|
| | Nav | Info | Mod | Nav | Info | Mod | |
| GPT-4O | 274 | 227 | 243 | 24 | 18 | 34 | 20 |
| CLAUDE-3.7-SONNET | 415 | 508 | 516 | 23 | 19 | 19 | 10 |

### B.1.2  NAVEXPLORER VS. PAGEEXPLORER TASKS

Since navigational tasks are important for website coverage and are linked to Go-Browse's outer loop, we also explicitly add a NavExplorer agent with Claude (chosen for its efficiency) in addition to the PageExplorer agents. This more than doubles the number of navigational tasks in the dataset.

Table 8: Comparison of Explorer types on navigation tasks.

| Explorer Type | # `Nav` Tasks | # `Nav` Task Clusters |
|---|---|---|
| NavExplorer | 925 | 32 |
| PageExplorer | 689 | 31 |

## B.2 DATASET COLLECTION COST ANALYSIS

Table 9 provides the cost per model during rollouts (both data collection and task proposal - Panel A) and also the cost of trajectory evaluation using GPT-4O (Panel B). The overall cost of collecting GO-BROWSE-WA is $975.57. We note that for trajectory rollouts, the cost of CLAUDE-3.7-SONNET, GPT-4O and GPT-4O-MINI is significantly reduced due to lower prices for cached tokens. We observe that ($\sim 53\%$ of input tokens are cache reads on average). We use the official OpenAI and Anthropic API pricing to compute their costs and use Together AI (Together AI) to estimate API cost for QWEN-2.5-7B-INSTRUCT.

Table 9: Go-Browse cost analysis for rollouts (agents) and trajectory evaluation.

| Panel A: Rollout Costs (Agents) | | | | |
|---|---|---|---|---|
| **Model** | **Num. Trajs** | **Num. Steps** | **Avg. Cost / Step** | **Total Cost ($)** |
| GPT-4o-mini | 11,695 | 95,314 | 0.0008 | 76.25 |
| Qwen-2.5-7B Instruct | 10,203 | 79,209 | 0.0025 | 198.02 |
| Claude-3.7-Sonnet | 5,102 | 24,532 | 0.0190 | 466.11 |
| GPT-4o | 103 | 789 | 0.0181 | 14.28 |
| **Total (Rollouts)** | **27,103** | **199,844** | **0.0037** | **754.66** |

| Panel B: Trajectory Evaluation Costs | | | |
|---|---|---|---|
| **Model** | **Num. Trajs** | **Avg. Cost / Traj Eval** | **Total Cost** |
| GPT-4O | 11,105 | 0.0199 | 220.91 |
| **Grand Total (Rollouts + Eval)** | | **$975.57** | |

## B.3 PAIRED BOOTSTRAP TEST FOR WEBARENA RESULTS

We measure statistical significance using the paired bootstrap test with 10,000 bootstrap samples of the WebArena benchmark results. Our model is statistically significantly better than QWEN-2.5-7B-INSTRUCT ($p < 0.001$). It was also judged as better than GPT-4O-MINI ($p = 0.108$) and NNETNAV-7B ($p = 0.094$). These results demonstrate a moderate degree of confidence in our model's improvement over these baselines, with high win ratios for GO-BROWSE-7B.

Table 10: Paired bootstrap tests of models vs. GO-BROWSE-7B on WebArena with 10K samples.

| Model Compared | Winner | Win Ratio (Baseline / Tie / GO-BROWSE-7B) | p-value |
|---|---|---|---|
| CLAUDE-3.7-SONNET | CLAUDE-3.7-SONNET | (1.000 / 0.000 / 0.000) | 0.000 |
| GPT-4O | GPT-4O | (1.000 / 0.000 / 0.000) | 0.000 |
| GPT-4O-MINI | GO-BROWSE-7B | (0.085 / 0.024 / 0.892) | 0.108 |
| NNETNAV-7B | GO-BROWSE-7B | (0.076 / 0.018 / 0.906) | 0.094 |
| QWEN-2.5-7B-INSTRUCT | GO-BROWSE-7B | (0.000 / 0.000 / 1.000) | 0.000 |

## B.4 OOD ANALYSIS ON ONLINE-MIND2WEB (OM2W)

In this section, we provide further analysis and discussion of the OOD OM2W experiment presented in Section 5. First, we use an LLM (GPT-4O-MINI) to classify each website in OM2W as being In-Domain-Adjacent (IDA) or truly Out-of-Domain (OOD) depending on how similar the OM2W tasks on these websites are to tasks on the WebArena websites. Fig 7 shows both the success rates on IDA and OOD websites on OM2W across the benchmarked models. We note the following interesting insights: (1) OM2W's IDA tasks seem to be significantly harder for models compared to OOD tasks, as can be seen by GPT-4O-MINI's much higher SR on OOD tasks compared to IDA tasks; (2) On IDA tasks, GO-BROWSE-7B performance is similar to GPT-4O-MINI performance (<1% difference in SR) while having a comfortable lead over NNETNAV-7B (3% difference in SR).

This suggests that models trained with GO-BROWSE continue to perform well on out-of-domain but similar websites to the ones explored during data collection.

Interestingly, NNETNAV-7B performs better than GO-BROWSE-7B on truly OOD tasks. To understand this better, and since OM2W provides difficulty labels per task, we also plot a breakdown of IDA and OOD success rates for the different models by task difficulty in Fig. 8. Here, we see that while NNETNAV-7B's OOD success rate increases over GO-BROWSE-7B's success rate on these tasks, most of the increase comes from better performance on easy tasks. Finally, in Table 11, we provide many examples of IAD and OOD succeses/failures for the different models spanning different difficulty levels.

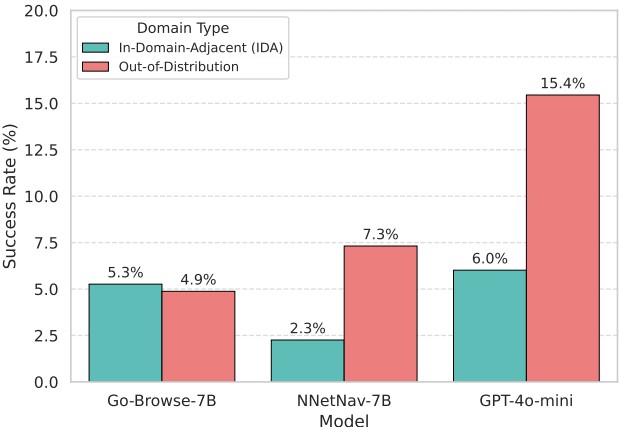

Figure 7: Breakdown of Success Rates of models on the Online-Mind2Web benchmark based on whether tasks come from In-Domain-Adjacent (IDA) or truly Out-of-Domain (OOD) websites.

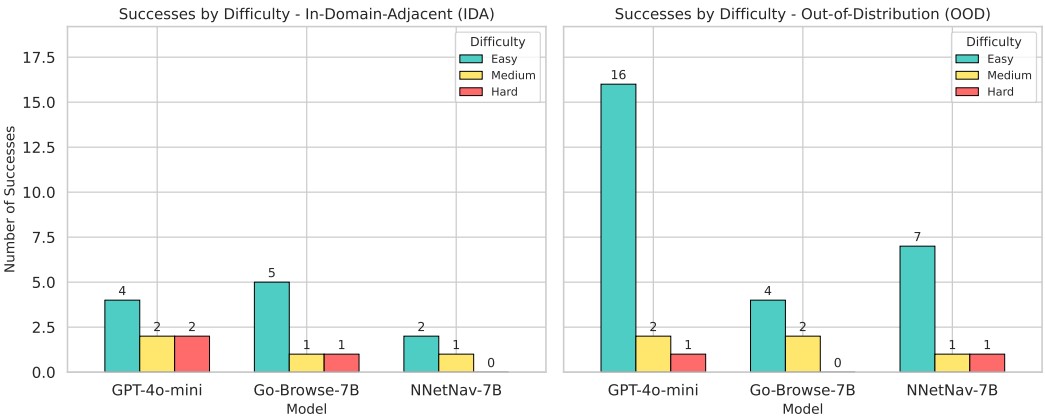

Figure 8: Breakdown of Success Rates of models on the Online-Mind2Web benchmark by both difficulty and whether the tasks are sourced from IDA or OOD websites.

Table 11: Example tasks from In-Domain-Adjacent (IDA) and Out-of-Distribution (OOD) website categories, showing successful (✓) and failed (✗) attempts. Difficulty: E=Easy, M=Medium, H=Hard.

| Model | Domain | Result | Diff. | Website | Task |
|---|---|---|---|---|---|
| Go-Browse-7B | IDA | ✓ | E | new.mta.info | Find the list of neighborhood maps for Brooklyn on new.mta.info. |

| Model | Domain | Result | Diff. | Website | Task |
|-------|--------|--------|-------|---------|------|
| Go-Browse-7B | IDA | ✓ | M | ign.com | Find a walkthrough for the game "The Legend of Zelda: Breath of the Wild" on ign. |
| Go-Browse-7B | IDA | ✓ | H | cvs.com | Find the most reviewed gluten-free multivitamins from CVS Health Brand under $15. |
| Go-Browse-7B | IDA | ✗ | E | google.com | Look for reviews of a Nest Hello Video Doorbell and filter by 1-star ratings. |
| Go-Browse-7B | IDA | ✗ | M | soundcloud.com | Browse a user homepage that reposted the top song from the Top 50 Rock chart. |
| Go-Browse-7B | IDA | ✗ | H | student.com | Find the lowest-priced Student housing near Liverpool International College which has been priced between 100 to 300 pounds and has a private bathroom. |
| Go-Browse-7B | OOD | ✓ | E | finance.yahoo.com | Show me historical data for EUR/USD. |
| Go-Browse-7B | OOD | ✓ | M | drugs.com | Find the Drug Interaction Report for Viagra and alcohol. |
| Go-Browse-7B | OOD | ✗ | E | qatarairways.com | Find the weight of baggage allowance for economy class on Qatar Airways. |
| Go-Browse-7B | OOD | ✗ | M | careers.walmart.com | Find support services jobs in Bentonville, in the state of Arkansas. |
| Go-Browse-7B | OOD | ✗ | H | healthgrades.com | Browse dermatologists within 10 miles of zip code 10019 and filter by only those who accept Blue Medicare Advantage. |
| NNetNav-7B | IDA | ✓ | E | allrecipes.com | Open the reviews of a recipe with beef sirloin. |
| NNetNav-7B | IDA | ✓ | M | ign.com | Find a walkthrough for the game "The Legend of Zelda: Breath of the Wild" on ign. |
| NNetNav-7B | IDA | ✗ | E | rottentomatoes.com | View the list of the Most Popular TV on rotten tomatoes. |
| NNetNav-7B | IDA | ✗ | M | akc.org | Find the nearest vet within 50 miles of zip 75228. |
| NNetNav-7B | IDA | ✗ | H | bbb.org | Get a quote from C and above-rated solar energy equipment company within 10 miles of Miami, Florida. |
| NNetNav-7B | OOD | ✓ | E | finance.yahoo.com | Find the closing stock price for Tesla on March 17, 2023. |
| NNetNav-7B | OOD | ✓ | M | careers.walmart.com | Find support services jobs in Bentonville, in the state of Arkansas. |
| NNetNav-7B | OOD | ✓ | H | chase.com | Using a calculator to determine how much I can have in my 401(k) account at retirement, if I work from age 22 to 65, with an annual rate of return of 3%, annual employee contributions of $8,000, and annual employer contributions of $8,000. |
| NNetNav-7B | OOD | ✗ | E | healthline.com | Find the recommended dosage for Vivitrol. |

| Model | Domain | Result | Diff. | Website | Task |
|---|---|---|---|---|---|
| NNetNav-7B | OOD | ✗ | M | petfinder.com | Find young cats in Seattle and show off the newest additions. |
| NNetNav-7B | OOD | ✗ | H | webmd.com | Search for the ovulation calculator and enter Mar 1 as the first date of the period and calculate the date of ovulation and pregnancy test day. |
| GPT-4o-mini | IDA | ✓ | E | eventbrite.com | Browse the page with event planning tips on Eventbrite. |
| GPT-4o-mini | IDA | ✓ | M | apple.com | Find the tech specs of the MacBook Pro 16-inch introduced in November 2023. |
| GPT-4o-mini | IDA | ✓ | H | leagueoflegends.com | Browse the final skin in the list for the champion Ahri. |
| GPT-4o-mini | IDA | ✗ | E | craigslist.org | Browse apartments with at least 2 bedrooms and 2 bathrooms and a max price of $4000 per month. |
| GPT-4o-mini | IDA | ✗ | M | usps.com | Find the nearest location to zip code 54620 that offers size 4 P.O. Boxes. |
| GPT-4o-mini | IDA | ✗ | H | airbnb.com | Find an Airbnb in Cleveland for three nights. The check-in date is the day after tomorrow. We have 2 adults, 2 kids, and 1 pet. The budget is $100 to $300 per night. Essential amenities include free parking, a washer, and a gym. |
| GPT-4o-mini | OOD | ✓ | E | drugs.com | Browse the natural products database. |
| GPT-4o-mini | OOD | ✓ | M | jobs.chronicle.com | Browse tenured/tenure-track faculty positions in Computer Sciences & Technology in California. |
| GPT-4o-mini | OOD | ✓ | H | chase.com | Using a calculator to determine how much I can have in my 401(k) account at retirement, if I work from age 22 to 65, with an annual rate of return of 3%, annual employee contributions of $8,000, and annual employer contributions of $8,000. |
| GPT-4o-mini | OOD | ✗ | E | thumbtack.com | Find a house cleaning service in 10001 on a weekly basis. |
| GPT-4o-mini | OOD | ✗ | M | sec.gov | Compare the U.S. ETP Odd Lot Rate (%) between Quartile 1 and Quartile 4, viewing quartiles by price, and display the chart with a logarithmic scale on the vertical axis. |
| GPT-4o-mini | OOD | ✗ | H | gov.uk | Check if a visa is required to work in the UK for longer than 6 months in Healthcare as an American citizen. |

## C   HYPERPARAMETERS AND ADDITIONAL EXPERIMENT DETAILS

For our finetuning experiments, we use the following hyperparameters. We train for 2 epochs on the whole dataset with a maximum sequence length of 24K tokens. We use a learning rate of 2e-5. We use a batch size of 8 (1 per gpu) with 4 gradient accumulation steps.

We used the following computational resources. For finetuning with a single NVIDIA 8xH100 node where each H100 has 80GB of VRAM. Training took ∼40 hours for each finetuning run. For dataset generation, we run on 5 nodes with of a SLURM cluster in parallel, with 256GB of RAM and 8 CPUs allocated to each, one each per WebArena domain. We also ran LLM inference servers on 8 NVIDIA L40S GPUs to support inferencing with QWEN-2.5-7B-INSTRUCT. Overall, dataset generation took ∼3 weeks to complete.

In this work, besides generating our own GO-BROWSE-WA dataset, we leverage the NNETNAV-WA dataset to build a baseline. This dataset was released with the Apache License 2.0 license.

## D   LLM USAGE

An integral part of GO-BROWSE is collecting data automatically by employing LLMs to explore and interact with websites. The usage of LLMs in this capacity has been detailed extensively in Section 3 and throughout the paper text. We also use LLMs to help with paper writing, particularly, to suggest revisions to phrasing of initial section drafts and to iterate on code for figures. All LLM generated revisions are further reviewed and revised by the authors before being included in the paper.

