# OpenReview forum: "Go-Browse: Training Web Agents with Structured Exploration"
_ICLR.cc/2026/Conference — ICLR 2026 Poster_

### Official Review · Reviewer_Hdmj · 2025-10-28

**Soundness:** 3
**Presentation:** 3
**Contribution:** 2
**Rating:** 4
**Confidence:** 4

**Summary:**

This paper proposes Go-Browse, a structured exploration framework for collecting high-quality interaction data for web agents. By treating the web as a graph and using an outer-loop frontier expansion with an inner-loop task discovery process, the method enables more efficient coverage of websites and deeper navigation compared to prior unsupervised approaches. Fine-tuning a 7B model on the collected dataset yields new SOTA results among sub-10B models on WebArena.

**Strengths:**

- Clear and intuitive idea: The graph-based formulation cleanly explains how and why structured exploration improves data efficiency, making the approach easy to understand and reproduce.

- Strong motivation with practical impact: Training web agents without human demonstrations is important and timely, and the proposed method directly contributes toward scalable data generation.

- Consistent empirical evidence: The paper presents solid improvements in both success rates and depth of exploration, demonstrating that the design choices (e.g., prefixed sampling) meaningfully contribute to better real-world web agent capabilities.

**Weaknesses:**

- Limited evaluation of generalization capability and lack of out-of-domain (OOD) analysis: In the Introduction (line 48-49), the authors argue that “agents are likely to be more successful if they learn directly from environments they will encounter.” While this claim is straightforward and somewhat obvious, it is impractical to train on all possible websites, and agents trained this way may be vulnerable to website updates. Therefore, generalization capability is crucial for web agents. However, the current paper’s analysis primarily focuses on in-domain performance, with insufficient analysis on OOD (Mind2Web) settings. It would be valuable to add additional results and analysis on OOD setups, such as (1) direct comparison with the baseline model and other methods, and (2) success and failure case analysis based on the similarity between in-domain training websites and OOD test websites

- Limited applicability of the proposed algorithm for capturing real-world web usage patterns: While the proposed graph-based algorithm improves coverage by systematically exploring web environments, it may not align well with how users interact with the web in practice. Specifically, the framework assumes a navigation-first, leaf-level task completion structure, where the agent traverses through a hierarchy of pages before executing an action. However, many realistic tasks (e.g., flight booking) require frequent alternation between navigation and value input across multiple intermediate pages. This discrepancy reflects a deeper human–agent misalignment, as the method prioritizes coverage over modeling the natural, goal-driven behavior of real users, potentially limiting its ability to capture the tasks that matter most in real-world web interactions.

- Lack of Analysis and Justification for VLM as a Judge:
The FeasibilityChecker component and the dataset filtering process seem to heavily rely on VLM as a judge, yet there is no analysis provided (e.g., robustness or correlation with human judges). Moreover, since the VLM judge evaluates success or failure based on screenshots, a justification is needed for whether this observation-based environment allows for accurate evaluations.

**Questions:**

See the weaknesses

---

> ### Author Response · Authors · 2025-11-21
>
> Thank you for your thoughtful feedback and highlighting the strong practical impact, intuitive approach and consistent empirical evidence of our approach.
>
> _**> Limited evaluation of generalization capability and lack of out-of-domain (OOD) analysis**_
>
> Thanks for the suggestion! We will add some more discussion and analysis on the OOD experiments in a revision and post this shortly.
>
> _**> Limited applicability of the proposed algorithm for capturing real-world web usage patterns**_
>
> We note that WebArena is a fairly realistic proxy for real websites. Firstly, the websites included in WebArena (such as GitLab, OpenStreetMap, Postmill, and others) are direct forks of their real-world counterparts (sharing the same source code), preserving most of their authentic complexity and functionality. These web sites are widely used in the real world, such as 50 million+ users who use GitLab.
>
> We also note that recent SOTA systems for web agent training also often take the approach of training on realistic and controllable web clones/sandboxes instead of on live websites. E.g., Yutori’s latest n1 web agent model was post-trained in Halluminate’s Westworld simulator [4].
>
> _**> Realistic tasks (e.g., flight booking) require frequent alternation between navigation and value input across multiple intermediate pages.**_
>
> Our exploration algorithm does not prevent tasks that require such back and forth, and if desired the models can easily be prompted during exploration and task proposal to add more tasks of this style. For instance, one of the proposed tasks included in the dataset is ‘View details of order ID 000000215 placed by Lucy Garcia."’ To solve this problem (especially in the unprefixed sampling case, where the agent starts from the website root/homepage), an agent may (1) navigate to the customers page (2) input some filters related to Lucy Garcia to try to find this order, (3) backtrack to the homepage once it does not find information on this page, (4) search for other relevant pages, (5) find and navigate to the orders page and (5) finally and filter and find the information on this page.
>
> _**> Lack of Analysis and Justification for VLM as a Judge**_
>
> We adopt an LLM-judge implementation similar to [1] which reports an evaluator accuracy of 80.6% when compared to ground-truth evaluation on WebArena. Note, we had an incorrect citation for this work in the original submission, and have now updated the citation in our paper revision (L277). This judge is also used by other works like [2, 3] to obtain strong performance on WebArena and other web benchmarks. So while some noise may be unavoidable with collecting data at scale, the LLM-judge  evaluator can still provide a good signal (as also demonstrated by our training results).
>
>
>
> **References**
>
> [1] Pan, Jiayi, et al. "Autonomous evaluation and refinement of digital agents." arXiv preprint arXiv:2404.06474 (2024).
> [2] Wang, Zora Zhiruo, et al. "Agent workflow memory." arXiv preprint arXiv:2409.07429 (2024).
> [3] Wang, Zora Zhiruo, et al. "Inducing programmatic skills for agentic tasks." COLM (2025).

---

> > ### Author Response · Authors · 2025-12-03
> >
> > Thanks once again for the suggestion to add more detailed discussion and analysis for the OOD experiments on Online-Mind2Web (OM2W). We wanted to follow-up to share that we have now added much more detailed results, analysis and examples for the OOD experiments to Appendix B.4 which we believe provides further evidence in favor of Go-Browse. We find that models trained with Go-Browse continue to perform well on out-of-domain but similar websites to the ones explored during data collection. Please see comment above "New Paper Revision: Additional analysis for OOD (Online-Mind2Web) Experiments and Real-World Website Performance." We hope this helps address your concerns regarding limited OOD analysis.

---

> ### Comment · Area_Chair_ohcP · 2025-11-25
> **Please participate in discussions with authors and other reviewers asap**
>
> Please ensure you are actively participating in the discussion with authors.
>
> Additionally, I strongly encourage you to read the other reviews and discuss with your fellow reviewers. It is vital to compare perspectives and raise any remaining concerns now to give the authors a fair opportunity to respond.
>
> Based on these interactions, please update your reviews and finalize your decisions.
>
> Best, AC

---

### Official Review · Reviewer_Bsi3 · 2025-10-31

**Soundness:** 1
**Presentation:** 3
**Contribution:** 2
**Rating:** 4
**Confidence:** 4

**Summary:**

Achieving effective exploration of web navigation agents can be challenging with existing approaches for the following reasons. Instruction-first approaches, which generate tasks (or task instructions) first and then navigate the web to solve it, are dependent on and limited by the prior information and knowledge about those web pages. On the other hand, interaction-first approaches, which navigate the web pages first and then post-process the collected trajectories, may be inefficient and less cost-effective. This work proposes a hybrid approch named Go-Browse. The authors view web navigation as expanding the graph of visited web pages and combine two types of explorations: global (outer loop) and local (inner loop) explorations. Local explorations are in-page explorations that happen in "frontier" nodes in the graph. With the use of vision-language models as judges, the tasks are used to gather trajectories and filtered. For scaling of the trajectory dataset, the authors perform additional sampling of trajectories on the filtered tasks by employing cheaper models to solve the tasks from the current web page (prefixed) or initial web page (unprefixed). Using the trajectory data collected in the WebArena environment, the authors fine-tune Qwen 2.5 7B and compare it with the baselines on Online-Mind2Web and WebArena to show the improved empirical performance.

**Strengths:**

1. Motivations
The work and proposed approach are reasonably motivated. As described by the authors, exploration in web environments is one of the challenges for web navigation data collection. Especially, the shortcomings of interaction-first and instruction-first approaches mentioned by the authors could be a bottleneck for scalable trajectory data collection on the web.

2. Presentation
The manuscript provides comprehensive information. It contains figures and pseudo-codes that make it easy to follow the content of the paper. Also, there are many statistics and analyses included, which can suggest insights about the empirical results from different angles. Additionally, the authors also care about reproducibility. The manuscript presents the prompts used for different components of the proposed approach. They also share a link to the source codes for their experiments. Overall, this can encourage the adoption and extension of this work.

**Weaknesses:**

1. Scalability of the proposed exploration approach
The authors use WebArena as the testbed for their exploration algorithm. However, the WebArena environment consists of concept/mockup websites and is limited in multiple aspects compared to real-world websites. Regardless of the number of unique web pages it provides, the structure of its websites and thus the possible patterns of navigation may not be diverse enough to test the scalability of the proposed approach. This is especially important, as exploration is more helpful and needed when there is more complexity in the environment.

2. Insufficient set of baselines
The authors compare the proposed method (Go-Browse) primarily with NNetNav, an interaction-first exploration method. On the other hand, while the authors mention instruction-first approaches as a relevant line of research, they do not perform an empirical comparison with such methods. Adding comparison with more baselines could make the submission stronger.

3. Comparison with NNetNav
While NNetNav is the only exploration approach that is being compared with Go-Browse, it does not look like an apples-to-apples comparison to me. There are some components that are reasonably fair for both: the same base model (Qwen2.5-7B-Instruct), data from the same WebArena environment, and comparable numbers of samples (around 39k steps for Go-Browse and around 45k steps for NNetNav). However, I am concerned about other components. Importantly, the authors make use of Claude Sonnet 3.7 and GPT-4o for different purposes. Figure 3 also states that the trajectory data from Claude Sonnet 3.7 constitute 33.9% of the final (successful) trajectory data. On the other hand, for NNetNav, Llama 3.1 70B Instruct was used. Therefore, the strength of the employed models could have meaningfully contributed to the current success rate wins vs. NNetNav on WebArena (2.9%) and Online-Mind2Web (1.33%).

**Questions:**

1. Is there empirical evidence of the applicability of Go-Browse to real websites with fair complexities? I believe this would be an important question for web navigation exploration approaches.

---

> ### Author Response · Authors · 2025-11-21
>
> We sincerely thank the reviewer for the thoughtful and detailed feedback and highlighting both the comprehensiveness of the manuscript and reproducibility of the work.
>
> _**> Is there empirical evidence of the applicability of Go-Browse to real websites with fair complexities?… WebArena environment consists of concept/mockup websites and is limited in multiple aspects compared to real-world websites.**_
>
> We disagree that WebArena is not a good proxy for real-world websites.
> The websites included in WebArena (such as GitLab, OpenStreetMap, Postmill, and others) are direct forks of their real-world counterparts (sharing the same source code), preserving most of their authentic complexity and functionality. Note that these web sites are widely used in the real world, such as 50 million+ users who use GitLab.
> Self-hosting the website lets us reset the state of each website for a fresh benchmark run and compare the agents fairly.
> As mentioned in Section 1, current models exhibit notably poor performance on WebArena and so improvements demonstrated on this benchmark are particularly significant and indicative of genuine advancements in agent capabilities.
>
> We also note that recent SOTA systems for web agent training also often train on realistic and controllable web clones/sandboxes instead of on live websites. E.g., Yutori’s latest n1 web agent model was post-trained using the Westworld simulator [1].
>
> _**> Insufficient set of baselines**_
>
> We chose NNetNav as our baseline as it was the previous SOTA for sub-10B parameter models on WebArena and importantly also released exploration trajectories on the WebArena benchmark that we could readily use to train on and compare against without needing to recollect data. Note that data collection is an expensive and time consuming process, often requiring weeks of work.
>
> Other methods instruction-first methods each collect data on different benchmarks and/or have implementation details that are not compatible. For example PAE [2] is an instruction-first method that is not fully autonomous (it first requires screenshots of human demonstrations as context to propose grounded tasks) and unlike our work or NNetNav’s, PAE uses VLMs instead of LLMs. Importantly its main experiments and data collection are on the WebVoyager benchmark, while WebArena experiments are more restricted (experiments on only 3/5 WebArena websites and evaluation on a custom easier WebArena-Easy benchmark they created). On this easier WebArena variant, PAE scores 23.1% SR with their 7B model. OS-Genesis [3] focuses on android/mobile GUI agents using (also with VLM-based agents instead of text-based LLM agents we use).  While not directly comparable for this reason, we note that OS-Genesis reports an accuracy of 10.79% on the full WebArena benchmark with Qwen2-VL-7B, which is significantly lower than Go-Browse-7B’s 21.7% SR. Explorer [4] is also a VLM based approach and performs exploration on URLs from similarweb.com and benchmarks on Mind2Web-Live and MiniWob++.
>
> We would also like to reemphasize that, our 7B parameter model trained on the Go-Browse-WA dataset is SOTA for sub-10B parameter models on WebArena and also beats more powerful general models like GPT-4o-mini.
>
> _**>  Comparison with NNetNav**_
>
> First, we acknowledge that there is a difference in models used in both methods. But note that we use a mixture of models which when compared to NNetNav’s usage of LLama-3.1-70B, range between much stronger (claude-3.7-sonnet), equivalent (gpt-4o-mini) and much weaker (qwen-2.5-7B-Instruct). In fact, as discussed in Section 6 - L414, a strength of the Go-Browse method is that it is able to sample a higher number of successful trajectories from weaker models via prefixed sampling.
>
> Table 9 of Appendix B performs a cost analysis showing that the cheaper models greatly amortize the cost of claude. But we use a strong model like claude-3.7-sonnet for the FeasibilityChecker module as the accuracy of this component greatly affects the efficiency of exploration (Section 6 - L430).
>
> We would also like to reemphasize that, independent of the comparison to NNetNav, our 7B parameter model trained on the Go-Browse-WA dataset is SOTA for sub-10B parameter models on WebArena, improves over the pretrained instruct model by 13.4% and also beats GPT-4o-mini.
>
> **References**
>
> [1] Halluminate. "Westworld: Benchmarking and Training Web Agents in Highly Realistic Simulators via Verifiable Rewards." Halluminate Blog (2025). https://halluminate.ai/blog/westworld
> [2] Zhou, Y. et al. “Proposer agent-evaluator (pae): Autonomous skill discovery for foundation model internet agents.” arXiv:2412.13194, 2024.
> [3] Sun, Qiushi, et al. "Os-genesis: Automating gui agent trajectory construction via reverse task synthesis." ACL. 2025.
> [4] Pahuja, Vardaan, et al. "Explorer: Scaling exploration-driven web trajectory synthesis for multimodal web agents." ACL 2025. 2025.

---

> ### Comment · Area_Chair_ohcP · 2025-11-25
> **Please participate in discussions with authors and other reviewers asap**
>
> Please ensure you are actively participating in the discussion with authors.
>
> Additionally, I strongly encourage you to read the other reviews and discuss with your fellow reviewers. It is vital to compare perspectives and raise any remaining concerns now to give the authors a fair opportunity to respond.
>
> Based on these interactions, please update your reviews and finalize your decisions.
>
> Best, AC

---

### Official Review · Reviewer_yijR · 2025-11-01

**Soundness:** 4
**Presentation:** 3
**Contribution:** 3
**Rating:** 4
**Confidence:** 5

**Summary:**

This paper proposes a pipeline for generating web trajectories via structured exploration that maintains a graph of discovered pages and resets to promising frontier nodes. A feasibility checker filters proposed tasks by attempting them with a strong model and using a VLM as a judge. Solvers then sample extra trajectories (prefixed/unprefixed) with cheaper models to scale data. The system is then instantiated on WebArena for collection and evaluation. Also evaluated on Online-Mind2web.

**Strengths:**

1. Treating websites as a URL graph with a maintained frontier reduces redundant exploration across episodes and helps reach deeper states that matter for task completion. The outer loop/frontier mechanism is well-motivated and is validated by broader site coverage and deeper success trajectories.
2. The dataset of ~10K trajectories is a valuable resource to the community to train web agents.

**Weaknesses:**

1. The evaluation results on Online-M2W are weak. While the authors say it is due to a different domain, it does not help sell their synthetic trajectory generation approach. The primary purpose of synthetic data generation for web agents is to improve their performance on real-world websites in the wild.
2. This proposed pipeline may not work as well on the real-world websites, as these are dynamic and the graph can change during the course of exploration.

**Questions:**

Is there a reason for choosing WebArena rather than the real-world web for trajectory synthesis? I would like to see results on Online-M2W in-domain after trajectory synthesis on those websites.

---

> ### Author Response · Authors · 2025-11-21
>
> We appreciate your feedback and your recognition of the soundness of approach and value of our released dataset to the broader community.
>
> _**> Is there a reason for choosing WebArena rather than the real-world web for trajectory synthesis? I would like to see results on Online-M2W in-domain after trajectory synthesis on those websites.**_
>
> We would like to note some reasons behind our choice of WebArena as the benchmark for our experiments:
>
> - **Diverse Web Domains**: WebArena includes five representative web domains—shopping, content management systems (CMS), GitLab, maps, and forums—providing a focused yet diverse set of realistic websites to study our setting of web agent specialization.
> - **Controlled Experimentation**: The benchmark websites in WebArena are self-hosted, enabling us to reset their databases to their original states after each agent interaction. This ensures uniformity and fairness in performance comparisons across different agent runs.
> - **Realistic Websites**: The websites included in WebArena (such as GitLab, OpenStreetMap, Postmill, and others) are direct forks of their real-world counterparts (sharing the same source code), preserving most of their authentic complexity and functionality.
> - **Challenging and Impactful Benchmark**: As mentioned in Section 1, current models exhibit notably poor performance on WebArena. Consequently, improvements demonstrated on this benchmark are particularly significant and indicative of genuine advancements in agent capabilities.
> - **Existing Framework Integration**: WebArena has previously been utilized effectively in prior research, including NNetNav, and is integrated into browser agent frameworks we build on top of (BrowserGym [2] & AgentLab).
> In addition, when we originally performed experiments for data collection with Go-Browse, OM2W had not yet been released. After it was released, we ran OOD experiments to get insights into how specialized models perform in the OOD website setting. Note, as described in Appendix B, data collection is time-consuming given our resources (on webarena, data collection took ~3 weeks to complete).
>
>
> _**> This proposed pipeline may not work as well on the real-world websites, as these are dynamic and the graph can change during the course of exploration.**_
>
> As described above, WebArena is fairly realistic — many of its websites use forks the source code of real websites that used by millions, such as the 50 million+ users of GitLab. models of today have still not saturated the benchmark. Regarding dynamic graph changes, note that the general structure of the graph changes for established websites is unlikely to change frequently (e.g., the general workflow of adding an item to a cart on Amazon or commenting on a post on reddit is likely to remain fairly stable). What can change frequently is the addition of new “leaf” nodes that are semantically similar to existing ones (e.g., new products added to amazon or new posts/comments on reddit). For such changes to the graph, Go-Browse should already capture workflow/task trajectories for many existing products/posts/comments on these websites. These should be similar to the kinds of tasks you would want to do on the new nodes.
>
> We also note that recent SOTA systems for web agent training also often take the approach of training on realistic and controllable web clones/sandboxes instead of on live websites. E.g., Yutori’s latest n1 web agent model was post-trained in Halluminate’s Westworld simulator [1].
>
> **References**
> [1] Halluminate. "Westworld: Benchmarking and Training Web Agents in Highly Realistic Simulators via Verifiable Rewards." Halluminate Blog (2025). https://halluminate.ai/blog/westworld

---

> > ### Comment · Reviewer_yijR · 2025-11-26
> >
> > Thanks a lot for your response. I am not fully convinced by the absence of experiments on Online-M2W (in terms of training data), especially since it was released in April 2025, many months before the submission deadline. The authors say WebArena has "diverse web domains," yet it does not transfer well to Online-M2W. However, W2 is party resolved, so I will update my score accordingly.

---

> > > ### Author Response · Authors · 2025-12-03
> > >
> > > Thank you for your constructive feedback and taking the time to reevaluate our submission and increasing your score.
> > >
> > > We wanted to also share that we have now added a much more detailed analysis in Appendix B.4, showing that for Online-Mind2Web websites that are In-Domain-Adjacent (not seen during training but still somewhat similar to WebArena websites), Go-Browse-7B has similar performance to GPT-4o-mini, suggesting that models trained with Go-Browse continue to perform well on out-of-domain but similar websites to the ones explored during data collection. See comment above on "New Paper Revision: Additional analysis for OOD (Online-Mind2Web) Experiments and Real-World Website Performance." We hope this helps address some of the remaining concerns about performance on OM2W.

---

> ### Comment · Area_Chair_ohcP · 2025-11-25
> **Please participate in discussions with authors and other reviewers asap**
>
> Please ensure you are actively participating in the discussion with authors.
>
> Additionally, I strongly encourage you to read the other reviews and discuss with your fellow reviewers. It is vital to compare perspectives and raise any remaining concerns now to give the authors a fair opportunity to respond.
>
> Based on these interactions, please update your reviews and finalize your decisions.
>
> Best, AC

---

### Official Review · Reviewer_Ur5A · 2025-11-01

**Soundness:** 3
**Presentation:** 3
**Contribution:** 3
**Rating:** 6
**Confidence:** 3

**Summary:**

This paper proposes Go-Browse, a go-explore inspired method for gathering training trajectories for web agents. Go-Browse leverages an outer loop that selects a previously discovered webpage, then performs an inner loop that uses modules to propose navigation and local tasks within a page, then uses verifier and task solver to gather trajectories for feasible tasks. Experimental results show that Go-Browse improves over the comparable state-of-the-art method for trajectory generation, Nnetnav, and analyses show that the proposed method results in deeper and wider exploration of the web environments.

**Strengths:**

- The proposed method is well motivated, and the adaptation of Go-Explore to exploring from a frontier of discovered webpages is intuitive and novel.
- The proposed method shows clear effectiveness over a comparable state-of-the-art method.
- The experiments and analyses are thorough, and all details as well as prompts are provided, enhancing reproducibility.
- The paper is well-written and easy to follow.

**Weaknesses:**

- The method leverages claude-3.7-sonnet for trajectory gathering, and it is unclear whether this may be a significant advantage of the proposed approach over NnetNav.
- I'm not sure I understand the purpose of the experiments on Online Mind2web, as the results seem to be evaluating WebArena-trained models on Online Mind2Web. However, my understanding is that the proposed method is more effective at exploring a given environment such as the websites in WebArena, while Online Mind2Web consists of entirely different websites.

**Questions:**

- Can you clarify "explore 20 different URLs for each of the five domains" (L269)? Does this mean that you use 20 starting URLs?
- What is the overall number of trajectories used for finetuning with Nnetnav (vs Go-Browse)?
- For finetuning, are both prefixed and unprefixed trajectories used together?
- What is the accuracy of the LLM judge? Is there any concern of label noise in the dataset from judge errors?

---

> ### Author Response · Authors · 2025-11-21
>
> Thank you for your review and highlighting the thorough analyses, reproducibility and clarity of the paper!
>
> _**> The method leverages claude-3.7-sonnet for trajectory gathering, and it is unclear whether this may be a significant advantage of the proposed approach over NnetNav.**_
>
> First, we acknowledge that there is a difference in models used in both methods. But note that we use a mixture of models which when compared to NNetNav’s usage of LLama-3.1-70B, range between much stronger (claude-3.7-sonnet), equivalent (gpt-4o-mini) and much weaker (qwen-2.5-7B-Instruct). In fact, as discussed in Section 6 - L414, a strength of the Go-Browse method is that it is able to sample a higher number of successful trajectories from weaker models via prefixed sampling.
>
> Table 9 of Appendix B performs a cost analysis showing that the cheaper models greatly amortize the cost of claude. But we use a strong model like claude-3.7-sonnet for the FeasibilityChecker module as the accuracy of this component greatly affects the efficiency of exploration (Section 6 - L430).
>
> We would also like to reemphasize that, independent of the comparison to NNetNav, our 7B parameter model trained on the Go-Browse-WA dataset is SOTA for sub-10B parameter models on WebArena, improves over the pretrained instruct model by 13.4% and also beats GPT-4o-mini.
>
> _**> I'm not sure I understand the purpose of the experiments on Online Mind2web, as the results seem to be evaluating WebArena-trained models on Online Mind2Web.**_
>
> Yes, the main focus of this paper is on specializing a model on websites of interest such. The WebArena evaluation captures this case. The OM2W experiments were performed to understand how specialized models perform on highly OOD scenarios (completely different websites). As we might expect, both of the specialized models (Go-Browse-7B and NNetnav-7B) perform poorly in this setting. Go-Browse-7B does however maintain a small lead over the NNetnav model.
>
>
> _**> Can you clarify "explore 20 different URLs for each of the five domains" (L269)? Does this mean that you use 20 starting URLs?**_
>
> Yes, this means we run the inner-loop (task proposal, feasibility checking and task-solving/trajectory sampling) starting from 20 URLs per website (i.e. outer-loop of Go-Browse is stopped after 20 iterations).
>
> _**> What is the overall number of trajectories used for finetuning with Nnetnav (vs Go-Browse)?**_
>
> NNetNav-WA contains 45K steps sourced from 4.7K trajectories. For Go-Browse, we train on 39K steps sourced from 9.5K trajectories. A reason for the difference in the number of trajectories even though the overall number of steps is comparable is that Go-Browse often separates/decouples navigation and page-local task solving into separate tasks/trajectories, leading to shorter trajectory segments, but still with high coverage of the website.
>
> _**> For finetuning, are both prefixed and unprefixed trajectories used together?**_
>
> Yes, for finetuning we use successful trajectories collected from both prefixed and unprefixed sampling.
>
> _**> What is the accuracy of the LLM judge? Is there any concern of label noise in the dataset from judge errors?**_
>
> We adopt an LLM-judge implementation similar to [1] which reports an evaluator accuracy of 80.6% when compared to ground-truth evaluation on WebArena. Note, we had an incorrect citation for this work in the original submission, and have now updated the citation in our paper revision (L277). This judge is also used by other works like [2, 3] to obtain strong performance on WebArena and other web benchmarks. So while some noise may be unavoidable with collecting data at scale, the LLM-judge evaluator can still provide a good signal (as also demonstrated by our training results).
>
> **References**
> [1] Pan, Jiayi, et al. "Autonomous evaluation and refinement of digital agents." arXiv preprint arXiv:2404.06474 (2024).
> [2] Wang, Zora Zhiruo, et al. "Agent workflow memory." arXiv preprint arXiv:2409.07429 (2024).
> [3] Wang, Zora Zhiruo, et al. "Inducing programmatic skills for agentic tasks." COLM (2025).

---

> ### Comment · Area_Chair_ohcP · 2025-11-25
> **Please participate in discussions with authors and other reviewers asap**
>
> Please ensure you are actively participating in the discussion with authors.
>
> Additionally, I strongly encourage you to read the other reviews and discuss with your fellow reviewers. It is vital to compare perspectives and raise any remaining concerns now to give the authors a fair opportunity to respond.
>
> Based on these interactions, please update your reviews and finalize your decisions.
>
> Best, AC

---

### Author Response · Authors · 2025-12-03

_**New Paper Revision: Additional analysis for OOD (Online-Mind2Web) Experiments and Real-World Website Performance**_

Thanks to Reviewer Hdmj once again for the suggestion to add more detailed discussion and analysis for the OOD experiments on Online-Mind2Web (OM2W). **We have now added much more detailed results, analysis and examples for the OOD experiments to Appendix B.4 which we believe provides further evidence in favor of Go-Browse.** Using LLM-based classification, we categorize websites in OM2W into one of (1) In-Domain-Adjacent (IDA) if its tasks in the OM2W benchmark are similar to tasks on WebArena websites or (2) Out-of-Domain (OOD) otherwise.

Fig. 7 and Fig. 8 in Appendix B.4 provide breakdown of success rates of models across OOD and IDA categories and also by task difficulty. A main takeaway from this analysis is that on tasks sourced from In-Domain-Adjacent websites, Go-Browse-7B performance is similar to GPT-4o-mini performance (<1% difference in SR) while having a comfortable lead over NNetNav-7B (3% difference in SR). This suggests that **models trained with Go-Browse continue to perform well on out-of-domain but similar websites to the ones explored during data collection. Given that Online-M2W websites are live websites, we believe this analysis also helps further address concerns from other reviewers regarding the applicability of Go-Browse to real-world websites.**

Table 11 further provides concrete examples of tasks across categories, difficulty level, and model success/failure.

---

### Meta-Review · Area_Chair_aTzs · 2026-01-11

**Summary:**

Reviewer Ur5A (score 6)
pointed out that using Claude to gather trajectories may not improve over existing similar benchmarks.

Reviewer yijR (score 4)
commented that the results on Online M2W were weak. There was also concern on how well this approach (designed for static graphs) would work on real-world websites with dynamic graphs.

Reviewer Bsi3 (score 4)
had concerns about the diversity of the graph underlying the websites in the benchmark, insufficient baselines and comparison to NNetNav (current approach uses Claude while the latter used LLama).

Reviewer Hdmj (score 4)
had concerns about limited evaluation on out of domain data, understanding differences between exploration performed by human subjects and the graph-based algorithm, and wanted more analysis on using the VLM as a judge.

I recommend that this paper be accepted. The reviews were somewhat brief and did not discuss the graph-based exploration approach in this paper at all. But the rebuttal has addressed them adequately.

**Reviewer Concerns:**

Reviewer Ur5A
The authors argued that using Claude seems to provide more successful trajectories (both in terms of the dataset, and also as evidenced by the fact that their model trained on Go Browse performs well on Web Arena). They have adequately addressed the questions

Reviewer yijR
The authors provided a number of reasons for favoring Web Arena (controlled experiments, works well with existing agents frameworks, etc.). They have tried to argue that websites on Web Arena are realistic because in many cases they use the same source code as the original websites. They showed that on websites from Online M2W that are similar to Web Arena, their model achieves a similar performance as that of GPT-4o.

Reviewer Bsi3
The authors have argued along similar lines as the response for Reviewer yijR. They argue that NNetNav was the previous state of the art on Web Arena. They have also provided some results in the Appendix to compare the cost of various models used in the analysis (e.g., Claude is more expensive but it significantly improves the efficiency of exploration).

Reviewer Hdmj
The authors provided more experiments in the Appendix on OOD data where models trained with Go Browser work well. They have pointed to examples in the dataset that reflect that the interactions discovered on Go Browse reflect real-world websites and usage patterns. This is anecdotal evidence rather than numerical analysis, but it is reasonable.

**Reviewer Scores:**

Reviewer Ur5A
I suspect that the score (6) would remain unchanged, it is pointing to an accept.

Reviewer yijR
The review said that they will update the score after the rebuttal. I am not sure whether the score in the system is the updated one. But I suspect that they are leaning towards an accept, e.g., a score of 6.

Reviewer Bsi3
The comments were addressed satisfactorily. I suspect the eventual score would have been a 6 (accept).

Reviewer Hdmj
The comments were addressed satisfactorily. I suspect the eventual score would have been a 6 (accept).

---

### Decision · Program_Chairs · 2026-01-26

Accept (Poster)